# Trunk Injection Delivery of Biocontrol Strains of *Trichoderma* spp. Effectively Suppresses Nut Rot by *Gnomoniopsis castaneae* in Chestnut (*Castanea sativa* Mill.)

**DOI:** 10.3390/biology13030143

**Published:** 2024-02-23

**Authors:** Alessandra Benigno, Chiara Aglietti, Santa Olga Cacciola, Salvatore Moricca

**Affiliations:** 1Department of Agricultural, Food, Environmental and Forestry Science and Technology (DAGRI), Plant Pathology and Entomology Section, University of Florence, Piazzale delle Cascine 28, 50144 Florence, Italy; chiara.aglietti@unifi.it (C.A.); salvatore.moricca@unifi.it (S.M.); 2Department of Agriculture, Food and Environment, University of Catania, 95123 Catania, Italy; sacaccio@unict.it

**Keywords:** *Castanea sativa*, nut rot, biocontrol, endotherapy, precision crop protection (PCP), sustainability

## Abstract

**Simple Summary:**

The cultivation of chestnut trees for fruit production has historically played a fundamental role in the survival of the populations of the poorest and most disadvantaged mountain areas of Southern Europe. Starting from the 2000s, a new fruit parasite, the fungus *Gnomoniopsis castaneae*, agent of the brown or chalky nut rot, has put chestnut cultivation in crisis. The control of this pathogen in the forest is difficult due to its endophytic lifestyle, wide distribution and the inability to resort to chemical control, given the need for environmental protection. In this study, strains of three *Trichoderma* species, *T. viride*, *T. harzianum* and *T. atroviride*, were tested for their ability to inhibit *G. castaneae*, both in the forest and in vitro. The inoculation of the antagonists was in the stem of adult chestnut trees using endotherapy, in two consecutive years. Statistically significant results demonstrated that the three biocontrol agents effectively suppressed nut rot in both chestnut stands and in vitro tests. Endotherapic treatments have proven to be an innovative and effective solution for the biological control of this emerging disease.

**Abstract:**

*Gnomoniopsis castaneae* is responsible for brown or chalky nut rot in sweet chestnut (*Castanea sativa*), causing heavy reductions in nut production. Controlling it is challenging, due to its inconspicuous infections, erratic colonization of host tissues and endophytic lifestyle. Fungicides are not applicable because they are prohibited in chestnut forests and strongly discouraged in fruit chestnut groves. *Trichoderma* species are safe and wide-spectrum biocontrol agents (BCAs), with a variety of beneficial effects in plant protection. This study tested selected strains of *T. viride*, *T. harzianum* and *T. atroviride* for their ability to suppress *G. castaneae*. Field experiments were conducted in four chestnut groves (two test plots plus two controls) at two sites with a different microclimate. As the size of the trees were a major drawback for uniform and effective treatments, the *Trichoderma* strains were delivered directly by trunk injection, using the BITE^®^ (Blade for Infusion in TrEes) endotherapic tool. The BCA application, repeated twice in two subsequent years, significantly reduced nut rot incidence, with a more marked, presumably cumulative, effect in the second year. Our data showed the tested *Trichoderma* strains retain great potential for the biological control of *G. castaneae* in chestnut groves. The exploitation of *Trichoderma* spp. as biopesticides is a novelty in the forestry sector and proves the benefits of these microbes in plant disease protection.

## 1. Introduction

Chestnuts are multi-purpose tree species, domesticated as crops since ancient times in various parts of the world, supplying nuts, timber, tannins and, indirectly, flour and honey [1]. Among these species, the sweet chestnut (*Castanea sativa* Mill.) has, for centuries, played a key role in many European countries as a primary source of livelihood for people living in mountainous areas [2].

In Italy, the second biggest producer of chestnut fruits in Europe (approximately 300,000 t in 2022 according to an FAO report) [3], the cultivation of sweet chestnut began to decline in coincidence with rural depopulation, an effect of the process of industrialization, which came with an unmanageable trend towards urbanization [4]. This abandonment of chestnut cultivation, essentially due to socio-economic development, was further accentuated starting from the period around World War II, this time due to biological factors, i.e., the advent of new diseases and pests. Among these are the introduction of the harmful chestnut blight pathogen *Cryphonectria parasitica* (Murr.) Barr.; the recrudescence of old infections by *Phytophthora cambivora* [(Petri) Buisman], the oomycete agent of “ink disease”; the more recent arrival of the cosmopolitan, plurivorous root rot pathogen *Phytophthora cinnamomi* Rands; as well as the introduction, in the last 20 years, of the Asian chestnut gall wasp *Dryocosmus kuriphilus* Yasumatsu. To make matters worse, biotic stresses were compounded by abiotic disorders, the major driver of which is climate change; climate warming, with high temperatures, low humidity and unusual weather events (e.g., increased recurrence of heat waves and drought during the growing seasons) cause severe defoliation and fruits abortion or non-ripening on chestnuts, with harmful consequences to chestnut cultivation and local economies [5]. 

The long-term crisis of chestnut cultivation, due to the depopulation of mountainous territories, the succession of parasitic attacks and climate anomalies [6] has shown, however, a marked reversal of trend in the last few years [7]. In recent years, in fact, chestnut cultivation has acquired a new and significant role, brought about by a new appreciation of the health and nutritional qualities of the fruit, as well as of the nutraceutical values of chestnut flours [8]. Furthermore, chestnut formations are being revalued enormously, due to a rediscovery of the range of ecosystem services these stands are able to offer, including their high landscape and cultural value, with an awakening of the demand from citizens for mountain traditions and environmental awareness [9]. 

Fruit spoilage by diseases, pests and abiotic factors was always one of the main causes of product loss in the chestnut production chain, but normally a low, tolerable percentage of fruits was affected in storage [10]. This situation changed drastically with the advent of a new fungal disease of the fruit. The disease was reported for the first time almost contemporarily in Australia and Northern Italy around 2012, with two different fungal species, now considered synonyms, which were recognized at that time as the causative agents: *Gnomoniopsis castaneae* Tamietti and *Gnomoniopsis smithogilvyi* L. A. Shuttleworth, E. C. Y. Liew, and D. I. Guest [11]. The fungus, belonging to the *Gnomoniaceae*, induces nut rot, with alterations in the internal tissue and organoleptic features of the fruit, which become white-brown in color, chalky and sponge-like in texture and takes on an unpleasant taste, becoming inedible. The heavy product losses caused by this pathogen today represent the most serious threat to chestnuts production [12]. Reports of the presence of this fungus are referred mainly to post-harvest investigations, with disease incidences that can vary depending on climate trend and environmental conditions of the growing areas but do not rarely reach 90% [13]. Besides causing nut rot in *Castanea* spp., this ascomycete was also reported to cause cankers on seedlings and young chestnut trees. It is also found as an endophyte in asymptomatic tissues of various tree and shrub species, including *Fraxinus ornus*, *Quercus ilex*, *Quercus cerris*, *Corylus avellana*, *Pinus pinaster* and *Buxus sempervirens* [14]. 

Due to the inconspicuous infections and the endophytic colonization of internal tree and nut tissue by *G. castaneae*, control measures against this harmful pathogen are, to date, limited and often ineffective [15]. Currently, most of the control methods rely on water curing, sterilization and the application of chemicals post-harvest [10,16]. However, the efficacy of these methods is strictly linked to disease severity, often being ineffective with high infection levels. The application of fungicides in chestnut stands, on the other hand, raises concerns for environmental problems, linked to their deleterious impact on non-target organisms and the emergence of pathogens’ resistant strains [6]. These effects, coupled with the risk chemical applications pose to human health, have led authorities to ban the use of pesticides in forest environments. Hence, alternative solutions that are effective against *G. castaneae* are highly demanded. 

The exploitation of the natural microbial enemies of plant pathogens as biocontrol agents (BCAs) is a new, promising strategy for sustainable crop protection, strongly promoted in Europe by national and EU agricultural and environmental policies [17]. Indeed, biocontrol enables the effective and environmentally friendly control of plant diseases by harnessing the natural ability of certain microorganisms to antagonize phytopathogens through various mechanisms. These mechanisms include competition for space and nutrients, direct parasitism, production of antimicrobial compounds and interaction with the plant system. This interaction enhances host growth and resistance by activating both ISR (induced systemic resistance) and SAR (systemic acquired resistance) pathways [18]. BCAs can be selected based on the targeted host–pathogen system among different organisms including fungi, bacteria and viruses that can be found in the ecosystem but among these, some genera (e.g., *Agrobacterium*, *Ampelomyces candida*, *Bacillus*, *Coniothyrium*, *Pseudomonas*, *Streptomyces* and *Trichoderma*) have acquired great importance, being marketed and promoted worldwide due to their BCA effectiveness [19]. Fungi of the genus *Trichoderma* have received much interest from both science and the commercial market, having been recognized as potential BCAs since the 1930s [20]. *Trichoderma* species are soilborne, free-living fungi that, depending on the strain and species, can effectively compete with naturally occurring microorganisms, restricting their growth for weeks or months. In many plant/pathogen/microbial antagonist interactions, these beneficial fungi revealed the ability to suppress pathogen growth in the rootsphere, endosphere and phyllosphere, either by direct mycoparasitism or indirectly, by releasing antifungal compounds (e.g., antibiotics) [21]. The use of *Trichoderma* species as BCAs, biostimulants and biofertilizers has become widespread in agricultural and horticultural systems, relying on different methods of application that range from seed coating to post-harvest and from soil to foliar treatments [22].

Contrary to a now widespread application in agriculture, biocontrol is still rarely applied in forestry, with the few attempts to apply BCA-based control referring mostly to lab, glasshouse or nursery investigations [23,24]. Schubert et al. [25] tested the use of *Trichoderma* spp. for controlling wood decay fungi in adult trees, obtaining effective results. However, the application of biocontrol in forests is particularly challenging due to the following several drawbacks: the steepness of the territory (for the effective execution of treatments with machinery), the absence of roads, and the large size of trees. All these difficulties can make a biopesticide treatment inconspicuous, erratic or incomplete, not to mention that some treatment types (e.g., foliar applications) would cause a large waste of the BCA due to drift and its dispersal into the environment, and this could negatively affect the ecological resilience of forest ecosystems, creating further damage [26]. The application of biocontrol agents by endotherapy could overcome some of these issues, by not provoking dispersal of the injected BCA into the environment and strongly reducing (practically zeroing out) the doses needed for treatment. The efficacy of endotherapic treatments coupled with the use of *Trichoderma* spp. has already been demonstrated by Berger et al. [27], who tested the method against *Phytophthora* species on *Quercus robur* and *Fagus sylvatica*. The aim of this study was to assess the effectiveness of some *Trichoderma* species for controlling *G. castaneae* rot in chestnut fruits, by delivering selected strains of these BCAs in cultivated *C. sativa* trees by trunk injections, using a minimally invasive technique. The test was repeated twice, in two consecutive growing seasons (2021 and 2022).

## 2. Materials and Methods

### 2.1. Study Sites

The study was conducted in chestnut groves at localities “Ribuio” and “Bandina”, in the Municipality of Ortignano-Raggiolo (AR), in the Casentino Valley (Arezzo, Tuscany), in central Italy. These localities were about 1300 m apart from each other, in two separate valleys with different microclimatic conditions (Ribuio test plot 43.683688 11.704197 690 S; Ribuio control plot 43.682456 11.705163 620 S; Bandina test plot 43.671514 11.700513 820 E; Bandina control plot 43.675462 11.705967 700 E (Figure 1). Two plots (ca. 0.4 Ha each), one per each locality, were established for the experimental trials. Two additional same-surface plots, homogeneous for tree cultivar and size, soil type and microclimate, were set up as a control in the immediate vicinity of each of the plots selected for treatments.

The “Ribuio” plot was on a skeleton in the form of outcropping rocks on a steep slope, in a small gorge, near a stand of *Pinus* sp. and was characterized by high humidity, due to the proximity of a river. It had a south-east exposure and an average altitude of 650 m a.s.l. (above sea level). The “Bandina” plot had no water course nearby, and the land was slightly sloping and was exposed to the east, with less outcropping rocks and fairly deep soil. The thermo-hygrometric conditions were therefore markedly different between the two plots (and in the adjacent control plots): “Ribuio” was more humid and colder; “Bandina” was hotter (because it was sunnier) and drier. The following parameters were recorded at each plot selected for treatments: number of trees, DBH (diameter at breast height: 1.30 m), height of each tree and age class (young < 30 years; adult 30–80 years; mature 80–150 years; overripe > 150 years height) (Table 1).

### 2.2. Soil Sampling and Trichoderma Isolation

A total of 20 soil samples were collected from the four study plots. Each area was divided into five homogeneous subplots, with one soil sample being collected from each subplot. Each sample was placed in a nylon bag, labeled and stored at 4 °C until use. For the isolation of *Trichoderma* from soil, modified malt extract agar (MEA, DIFCO, Detroit, MI, USA) was used. Six-fold serial dilutions of each soil sample were then prepared in sterilized, distilled water and 1 mL of diluted sample was poured onto the surface of the MEA, amended with 25 mg/L of streptomycin sulfate (Sigma Aldrich, Steinheim, Germany). The plates were incubated in the dark at 24 ± 1 °C for 7 days, according to the species requirements. Morphologically different colonies appearing on the plates were sub-cultured onto MEA and stored at 4 °C.

### 2.3. Identification of G. castaneae and Trichoderma *spp.* Strains

*G. castaneae* strains were isolated from infected nuts, whereas *T. viride* and *T. harzianum* strains were retrieved from soil samples. The appearance and morphology of the colonies (surface topography, texture, compactness, mycelium pigmentation and margin type) were determined on malt extract agar (MEA). For each morphotype, tufts of mycelium were picked off by scraping colony surfaces with a sterile dissecting needle, then mounted on glass slides in a drop of 0.5% KOH or Lactophenol for direct examination under the light microscope. Conidial micromorphology was determined under a Zeiss light microscope (ZEISS, Jena, Germany) at ×40 magnifications, by averaging 200 measurements per fungal taxon. Images were captured with an Optikam 4083.B5 microscopic Digital USBCamera operated with OptikaView version 7 acquisition software (OptikaSrl, Ponteranica, Italy). The identity of representative strains of each species was confirmed through sequence analysis of the region spanning the 5.8S rRNA gene and flanking Internal Transcribed Spacers 1 and 2 (ITS1-5.8 S-ITS2). For DNA extraction, fungal strains were grown on sterile cellophane in 90 mm Petri dishes containing 1% potato dextrose agar (PDA, DIFCO, Detroit, MI, USA) and maintained in the dark at 20 °C. After 7 days of incubation, ca. 70 mg (fresh weight) of mycelium was scraped off the cellophane surface and stored at −20 °C in a 2 mL Eppendorf tube until use. Genomic DNA was extracted using the GenElute plant Genomic DNA Miniprep extraction kit (Sigma Aldrich, St. Louis, MO, USA), following the manufacturer’s instructions, and stored at −20 °C. Internal transcribed spacer (ITS) region PCR amplification was performed on extracted samples by using ITS1 (5′-TCC GTA GGT GAA CCT GCG G-3′) and ITS4 (5′-TCC TCC GCT TAT TGA TAT GC-3′) universal primers [28]. Cycling and sequencing conditions were as described in Moricca et al. [29]. The identification of retrieved fungal taxa was undertaken by processing the relative sequences with the nucleotide–nucleotide BLAST^®^ search tool (Basic Local Alignment Search Tool; http://www.ncbi.nlm.nih.gov/BLAST accessed on 11 October 2023). Generated sequences were submitted and deposited in NCBI GenBank database.

### 2.4. In Vitro Antagonism Tests

The *T. harzianum* and *T. viride* isolates retrieved from the soil were confronted with *G. castaneae* for their ability to inhibit the pathogen’s mycelial growth (%) in in vitro dual culture assays. For comparison, the test also included a commercial strain (SC1) of *T. atroviride* obtained from a Vintec^®^ biological commercial formula (Certis Belchim B.V., Utrecht, The Netherlands). Each test was performed in a 9 cm diameter Petri dish containing 12 mL of MEA. Two plugs (diameter, 4 mm) retrieved from 7-day-old colonies of *G. castaneae* and of each one of the selected antagonists, were inserted in each dish, 6 cm apart from each other. Mycelial interactions were determined according to Badalayan et al. [30]. The experiment was conducted with 10 replicates for each antagonist and for the control, represented by two isolates of *G. castaneae*. Dishes were incubated at 25 °C in the dark for 8 days. Mycelial radial growth was measured after 8 days in each dish, calculating the growth area of each fungal species using the ImageJ software (version 1.8.0). Results were then compared to those obtained for the controls. The inhibition index percentage (I%) was calculated as follows [31]:$$I\%=\left[\frac{\left(RM-rm\right)}{RM}\right]\times 100$$
where *RM* is the radius of the *G. castaneae* colony in the control plate, and *rm* is the radius of the colonies in the direction of the antagonist.

Hyphal interactions between colonies were assessed at 7 and 14 days under an optical microscope (ZEISS, Jena, Germany), with the antagonistic ability of *T. viride*, *T. harzianum* and *T. atroviride* specifically determined following Badalayan et al. [30], using the rating scale described in Table 2.

### 2.5. Biocontrol Agent Mixture Preparation

The solution used for endotherapic treatments was prepared by collecting *T. harzianum* and *T. viride* conidia from pure 7–10-day-old MEA cultures. For *T. atroviride*, the preparation was made according to the manufacturer’s instructions for the Vintec^®^ (Belchim Crop Protection, Londerzeel, Belgium) biological formulation. The collected fungal mixture was filtered through filter paper (Whatman^®^ Grade 1, Buckinghamshire, UK, diameter: 9 cm, pore size: 11 µm) to remove mycelium fragments. The concentration of conidial suspensions was then adjusted to 10^8^ conidia/mL of water by using a counting chamber. The final dose of the fungal mixture utilized in trunk injections was 0.8 mL/10 cm of trunk circumference, following Berger et al. [27].

### 2.6. In Vivo Tests

In order to investigate the ability of *Trichoderma* species to antagonize the nut rot agent *G. castaneae* in planta, selected strains of the BCAs were delivered by means of trunk injections into the 66 adult chestnut trees growing in the two chestnut groves Bandina (34 trees) and Ribuio (32 trees) (see Table 1 above). Endotherapic treatments were performed at breast height (about 1.5/1.7 m from the ground) by using the BITE^®^ (Blade for Infusion in TrEes) injection tool from De Rebus Plantarum (Vicenza, Italy). This instrument enables a targeted delivery of the biopesticide, without producing holes in the tree, but rather by causing a minimal (a few mm) vertical lesion, which the plant normally heals in a couple of weeks [32]. The *Trichoderma* solution was applied in June, when the trees were actively growing, with the crown fully expanded, and the full hydraulic tension of the transpiration flow guaranteed the maximum absorption of the solution (Figure 2). Treatments were performed during two subsequent growing seasons, 2020 and 2021.

### 2.7. Assessments of G. castaneae Incidence

Plants were monitored for their health conditions throughout the duration of the field experiment, with plant material (foliage, shoots, green curls with unripe nuts) being collected and analyzed periodically. Up to 400 fully ripened nuts were finally randomly collected from chestnut trees in each treated and untreated stand in October (Figure 3a,b). Fresh chestnuts were transported to the laboratory and stored at 4 °C before isolations. Each chestnut was surface washed with 75% ethanol (1 min) and 3% sodium hypochlorite (NaOCl) (3 min), then rinsed three times in sterile water. A sterile scalpel was used to remove the outer lignified shell and open the nut by cutting it in half. Five fragments (approximately 1 × 1 × 2 mm in size) were randomly excised from the tissues of each fruit and plated in 90 mm diameter Petri dishes filled with 2% MEA. All dishes were incubated in the dark at 25 °C for 3 days, according to species requirements. The presence of *G. castaneae* was analyzed for each dish by comparing macro- and micro-morphological features of each obtained mycelium with that of a *G. castaneae* isolate, whose identity was previously assessed by ITS sequencing (Figure 3c).

### 2.8. Statistical Analyses

For all the traits considered, the means and standard deviations were calculated for all the combinations considered. A one-way ANOVA was performed to test the significance of the in vitro antagonistic activity of three *Trichoderma* spp. versus *G. castaneae*. All data refer to the four chestnut stands (two treated and two untreated). Homogeneity of variance and normality tests were performed using the Levene and Shapiro–Wilk tests. SPSS V.28 (IBM Corporate, Endicott, NY, USA) was used for the statistical analysis, with χ^2^ square test, to identify significant differences among treatments. The square test was applied to analyze the significance of the differences between trees treated with *Trichoderma* spp. and the controls (with no BCA application). The incidence of *G. castaneae* was calculated for each treated and untreated stand as the ratio (%) between the number of affected nuts and the total number of nuts analyzed.

## 3. Results

### 3.1. Gnomoniopsis castaneae and Trichoderma *spp.* Identification

The colonies of *G. castaneae* were initially white, turning to light gray after 7 days of incubation. The mycelium appeared either flat or thick and densely woolly. Margins were diffuse and developed in concentric circles. In the innermost portion of the colony, acervuli, ranging in color from orange to red, developed after 10 days. They were either superficial or erupting on the upper surface, circular, solitary or gregarious. Conidia were hyaline, one-celled, ovoid–oblong, straight or curved. Colony phenotypes, as well the size and shape of conidia, matched exactly the original descriptions of Visentin et al. [33] and Shuttleworth et al. [34]. DNA sequencing confirmed the representative morphotype as belonging to *G. castaneae* (PP326312), with BLAST searches that revealed complete (100%) sequence homology with those of the pathogen already deposited in the GenBank database. *Trichoderma harzianum* exhibited a cottony white, slightly yellowish mycelium, with a flat growth profile. Colonies developed at 8 days showed several concentric rings with dark green conidial production. *Trichoderma viride* colonies showed a uniform appearance with light green-yellowish conidia evenly distributed across their surfaces (without differentiating concentric rings). Growth rates and cultural characteristics of the two *Trichoderma* species differed on the same medium at a constant temperature of 24 °C. *T. viride*, from all soil types, demonstrated the highest growth rate compared to *T. harzianum*. *Trichoderma harzianum* conidia were from globose to sub-globose, with a dark green color. *T. viride* presented globose conidia, with a color ranging from light yellow to green.

Sequence analysis confirmed the identities of *T. harzianum* and *T. viride* (GenBank acc. nos. PP326311 and PP326313, respectively), with BLAST searches that revealed complete (100%) homology with sequences of the two species retrieved from the GenBank database.

### 3.2. In Vitro Tests

A priori in vitro assays were useful for assessing *Trichoderma* species for their ability to inhibit *G. castaneae*. *T. viride* and *T. harzianum* strains were highly effective against *G. castaneae*. The parasitization of the pathogen was already visible at 96 h. After 8 days, *G. castaneae* mycelium turned out completely overgrown and replaced by strains of these two *Trichoderma* species (Figure 4a,b). The growth of *T. atroviride* at 25 °C was lower than that of *T. viride* and *T. harzianum*, with only a partial interaction with *G. castaneae* that was observed at the margin of the two colonies (Figure 4c). Hyphal interactions for *T. viride* and *T. harzianum* resulted in the CA2 subtype of Badalyan’s scale; *T. atroviride* ranked as CA1 of the same scale. Statistical analyses confirmed the different behaviors/inhibitory effects of the three *Trichoderma* species. The one-way ANOVA statistical criterion showed that *T. atroviride*’s area value differed significantly after 6 days from that of the other two antagonistic isolates (*p* > 0.05). The highest inhibition capability was found for *T. harzianum* (99.14%), followed by *T. viride* (78.12%) and *T. atroviride* (51.82%) (Table 3).

### 3.3. Endotherapic Treatments in the Field

The field trials proved the effectiveness of treatments with *T. viride*, *T. harzianum* and *T. atroviride* in curtailing *G. castaneae*, with the pathogen’s incidence being significantly reduced in the investigated chestnut groves. Data from the first and second year of treatments showed a reduction in *G. castaneae* incidence, with this trend increasing in the second year. In fact, in 2020, different percentages of *G. castaneae* incidence were obtained in the two treated chestnut groves, which were 24% and 22%, respectively, compared to the two control areas (46% and 48%). In the second year of treatment (2021), the incidence of the target pathogen was 23% and 11% in the treated plots and 63% and 40% in the in the control plots. The decrease in *G. castaneae* incidence between treated and untreated plots resulted in percentages of 26% and 22% in 2020 and of 40% and 29% in 2021 (Figure 5). The chi-squared statistical analysis confirmed the significant differences (*p* ≤ 0.01) between treated and untreated chestnut groves during the two years of analysis. No other nut rot pathogens were isolated from the analyzed nut samples.

## 4. Discussion

Nut rot by *G. castaneae* today represents the main cause of chestnut fruit deterioration. The disease strongly curtails nut production in the areas where it is present, the pathogen irremediably compromising nut quality and organoleptic features, making the fruit tissue chalky and inedible [11,13,35,36,37,38,39]. As the fungus can live as an endophyte in latency in chestnut organs and tissues, the application of effective control measures is troublesome and relies only on post-harvest treatments: water curing, sterilization and chemicals application [10,16]. Since a chestnut grove is considered in all respects a forest, chemical treatments are prohibited for environmental reasons. But even if they were not prohibited, chemical treatments would still be difficult to implement due to the morphology and slope of the terrain on which chestnut groves are often located, the environmental and spatial heterogeneity within the plantation and the large size of the trees. For all these reasons, in this work, biological control based on the application of *Trichoderma* species was evaluated as an alternative strategy for the control of *G. castaneae* in chestnut groves. The implementation of biological control against plant pathogens in agro-ecosystems by releasing competent *Trichoderma* strains is a safe technology that could solve issues related to environmental and health risks inherent with conventional control strategies [40,41]. Fungi of the genus *Trichoderma* are innate with the environment, being common inhabitants of soils, both agricultural and forest, and are often also found as endophytes in many plant species [42]. The effectiveness of *Trichoderma* species as BCAs against plant pathogens has been known for a long time and has been successfully applied in a variety of agricultural crops, especially in horticulture [43,44,45,46,47,48,49].

The application of biological control methods against forest pathogens, on the contrary, has been scarcely investigated and few attempts were made on adult trees [25,26]. In this work, endotherapy was exploited to deliver *Trichoderma* BCAs against *G. castaneae* directly in the stem of cultivated chestnut trees. Results were promising and consistent with those obtained by Berger et al. [27], who injected *T. atroviride* in *Quercus robur* and *Fagus sylvatica* using endotherapy as a preventive strategy against *Phytophthora* spp. infections. Indeed, our results showed that, in 2020, the incidences of *G. castaneae* were 24% and 22% in the treated groves (Ribuio and Bandina), in comparison to their respective adjacent control plots (46% and 48%). A similar behavior was observed in the second year of treatment (2021), registering pathogen incidences of 23% and 11% in the treated plots (Ribuio and Bandina), and 63% and 40% in the control plots (Ribuio and Bandina). The decrease in *G. castaneae* incidence between treated and untreated stands resulted in percentages of 22% and 26% for 2020, and of 29% and 40% for 2021. The difference between the two locations, in the incidence of *G. castaneae* both in treated and in control plots, is ascribed to the different thermo-hygrometric conditions of the two sites. In fact, Ribuio is characterized by a particular microclimate: it is in a gorge, at the base of which a stream flows, which generates constant humidity and poor sunlight on the ground. Bandina, on the other hand, has a different exposure, is not located near streams and experiences high levels of sunshine. The particular temperature and humidity conditions of Ribuio may have favored *G. castaneae* infections.

The results of in vitro tests performed in this study have further confirmed the effectiveness of *Trichoderma* application as a biocontrol agent, as it inhibited *G. castaneae* in culture. Indeed, the inhibition of *G. castaneae* was observed by each tested *Trichoderma* species, with inhibition index percentages of 99.14% for *T. harzianum*, 78.12% for *T. viride* and 51.82% for *T. atroviride*. To the best of our knowledge, no other authors have reported a detailed assessment of the effects and interactions among *T. viride*, *T. harzianum* and *T. atroviride* on *G. castaneae* using dual culturing. Our in vitro results are in line with those of other authors who tested these *Trichoderma* species against different plant pathogens [24,50,51].

The treatment with three *Trichoderma* spp. was able to significantly reduce the necrotic surfaces of chestnuts caused by *G. castaneae*, and no systematic studies of resistance have been undertaken to date [11]. The results obtained in this study can also be compared with those reported by Pasche et al. [23], who applied *T. atroviride* against *C. parasitica* and *G. castaneae* by soaking chestnut scions in fungal propagule suspensions. These authors observed that the endophytic behavior of *Trichoderma* in chestnut tissues and the presence of *T. atroviride* were able to influence the infection of *G. castaneae*. Indeed, in agreement with our study, only 15% of treated scions analyzed by Pasche et al. [23] showed *G. castaneae* symptoms*,* while the percentage of *G. castaneae* symptoms in control scions was 75%. These authors ascribed the effectiveness of their biological control treatment to the possibility that inoculated *Trichoderma* had spread and colonized the totality of woody tissues.

The endotherapic method applied in this work in chestnut trees could improve the efficacy of biological applications, facilitating the entrance of *Trichoderma* in the plant vascular system by trunk injection. Endotherapy offers important advantages over other treatment methods: it lowers (reducing practically to zero) the dispersion of the BCA into the environment, and the doses needed for treatments are enormously reduced, making them more sustainable than traditional methods of application (e.g., spraying). The results obtained in this study suggest that the biological control method employed could represent a key advantage for chestnut growers, since it reduces the decay of fresh and processed nuts and the resulting economic losses.

Little is known about the duration of protection conferred by treatment with *Trichoderma* BCAs in forestry. However, the fact that in the second year a greater reduction in the incidence of rot was achieved means that there was an additive effect between the first and second year treatments. The mechanisms of action of *Trichoderma* strains (e.g., direct parasitism, lytic enzyme production, antibiosis, competition for nutrients and space) have been widely studied [18,19,21,22,50,52,53,54]. It is possible that the injected *Trichoderma* species utilized one or more of the above mechanisms, as well as also synthesizing bioactive compounds that elicited plant defense molecular responses, through the activation of either the ISR (induced systemic resistance) or the SAR (systemic acquired resistance) pathways. All these protection mechanisms may have restricted *G. castaneae*, conferring long-lasting defense to chestnut trees [18,41].

## 5. Conclusions

This is the first work in which BCAs of the genus *Trichoderma* have been administered on chestnut trees by endotherapy. The choice of endotherapic applications, dictated by technical–operational needs (we needed to administer the biopesticide in uncomfortable conditions, on large-sized plants and, last but not least, to reach the target pathogen in the most distal portions of the foliage), turned out to be a valid option. Further research is required to confirm the promising results obtained here and to better elucidate unknown aspects of *Trichoderma* mycoparasitism when conferring these BCAs to restrict *G. castaneae* in chestnut groves.

## Figures and Tables

**Figure 1 biology-13-00143-f001:**
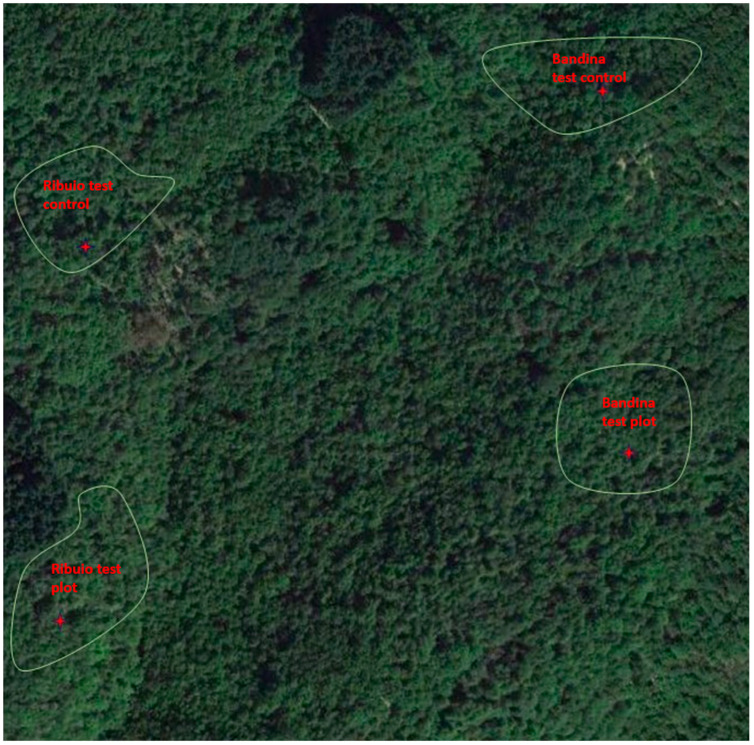
Aerial view of the four plots (Ribuio test, Ribuio control, Bandina test and Bandina control); the plots under investigation are delineated in green.

**Figure 2 biology-13-00143-f002:**
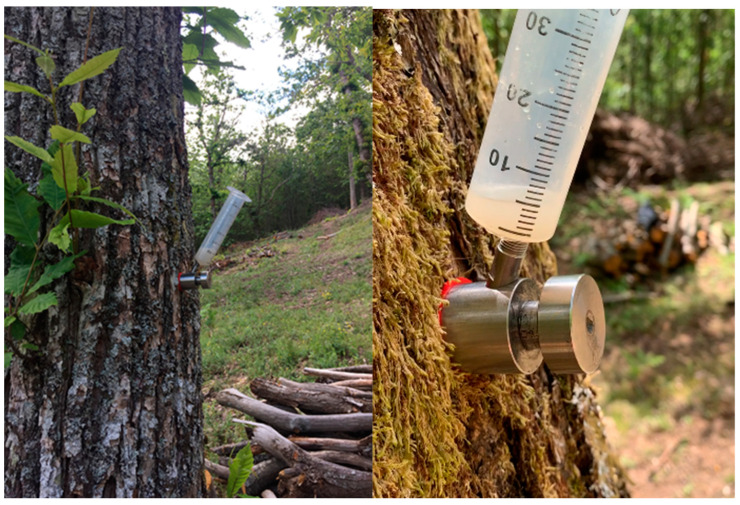
Application of conidial suspensions of the BCAs on *C. sativa* stems with the BITE^®^ injection tool.

**Figure 3 biology-13-00143-f003:**
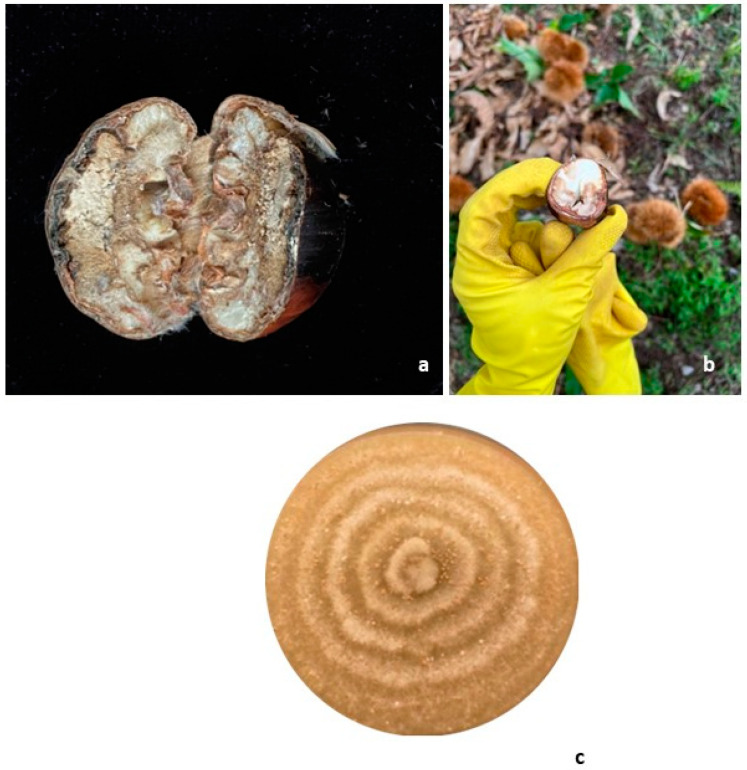
Symptoms caused by *G. castaneae* in nuts: (**a**) a fruit with a completely dehydrated endosperm; (**b**) a freshly collected nut and cut in the field displaying a spongy or chalky appearance with distinctive dark-brown lesions. This discoloration further underscores the altered composition, indicating potential physiological changes within the fruit; (**c**) a *G. castaneae* culture grown on malt extract agar (MEA) after two weeks.

**Figure 4 biology-13-00143-f004:**
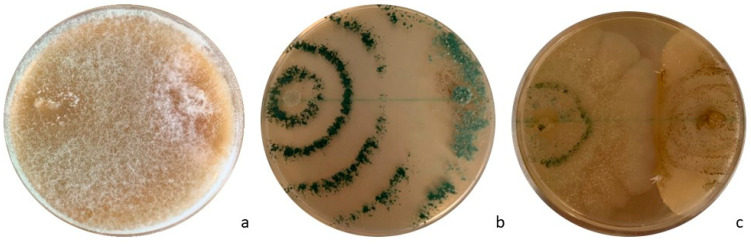
In vitro dual cultures of *Trichoderma* species against *G. castaneae* after two weeks of incubation at 25 °C: (**a**) *T. viride*/*G. castaneae* interaction; (**b**) *T. harzianum*/*G. castaneae* interaction; and (**c**) *T. atroviride*/*G. castaneae* interaction.

**Figure 5 biology-13-00143-f005:**
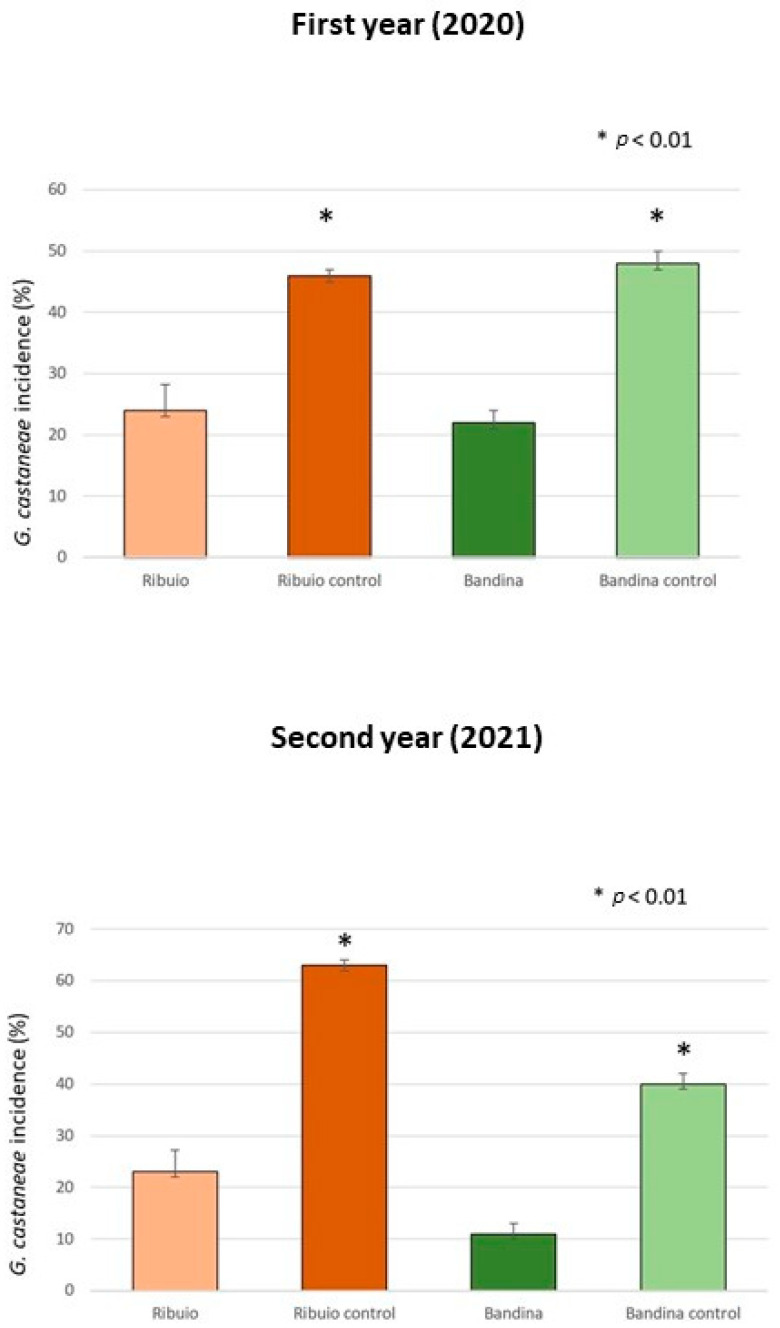
Average percentage of nut fruits found infected by *G. castaneae* in test plots at the “Ribuio” and “Bandina” chestnut groves and in their respective control plots. Black bars represent standard errors. * Significant differences between numbers of infected nuts found between localities (χ^2^, *p* < 0.01).

**Table 1 biology-13-00143-t001:** Age class, DBH (cm) and height (m) of treated trees in each chestnut stand.

Ribuio Chestnut Stand			
Number	Age (Y = Young, <30 years; M = Mature, 80–150 years; O = Overripe, >150 years)	Diameter at Breast Height DBH (cm)	Height (m)
1	O	79.58	11.8
2	O	42.97	15.9
3	O	86.26	14
4	O	95.49	12.3
5	O	44.56	10
6	M	38.2	14.7
7	M	34.06	13.8
8	O	113.32	12.8
9	M	53.16	16.3
10	M	33.42	14.1
11	O	65.57	10.9
12	O	108.23	10.9
13	O	113	11.1
14	O	66.85	9.4
15	O	82.76	7.6
16	O	64.3	7.8
17	O	43.93	12.8
18	O	69.39	7.7
19	M	36.61	6.1
20	NA	28.87	NA
21	O	62.71	6.9
22	O	70.66	6.8
23	O	60.48	NA
24	NA	60.48	NA
25	O	63.66	NA
26	NA	49.34	NA
27	O	92.31	NA
28	O	98.04	NA
29	O	127.32	NA
30	M	49.34	NA
31	O	119.37	NA
32	O	93.9	NA
Bandina chestnut stand			
1	M	70.03	10
2	M	38.83	11.9
3	M	55.7	10.02
4	M	53.16	11.06
5	M	59.21	13.6
6	M	41.38	13.4
7	M	61.75	15.1
8	M	51.57	14.1
9	M	45.2	14
10	M	48.38	16.1
11	M	74.8	9.5
12	M	44.56	12.9
13	M	67.16	10.5
14	M	73.21	11.1
15	M	49.34	16.8
16	M	53.48	16
17	M	46.47	11.4
18	M	48.38	12.2
19	M	59.84	14
20	M	60.8	11.9
21	M	50.93	11.5
22	M	66.85	11.5
23	M	58.89	10.3
24	O	194.17	10.9
25	Y	30.24	13.6
26	Y	35.01	13.4
27	M	48.7	13.5
28	M	49.02	12.4
29	Y	29.92	16.3
30	M	59.21	14.2
31	M	37.24	14.5
32	M	42.34	14.7
33	M	40.11	14.3
34	M	61.75	15.6

NA = Not Available.

**Table 2 biology-13-00143-t002:** Rating scale of the interaction types among the antagonistic fungus and investigated pathogen, following Badalayan et al. [30].

Type of Interaction	Interaction	Value
A	Stop of colony growth by contact with mutual inhibition	1
B	Remote stop without mycelial contact	2
C	Growth of one colony over another without initial stop	3
CA1	Partial growth of one colony over another after contact arrest	3.5
CA2	Complete growth of one colony upon another after contact arrest	4.5
CB1	Partial growth of one colony upon another after remote arrest	4
CB2	Complete growth of one colony upon another after remote arrest	5

**Table 3 biology-13-00143-t003:** Antagonism of *T. viride*, *T. harzianum* and *T. atroviride* against *G. castaneae* assessed after 6 days by (A) average *Trichoderma* spp. area; (I) inhibition index percentage of *G. castaneae* mycelial growth; and (CI) competitive interactions.

Isolates	A (cm^2^) *	I (%) *	CI **
*Trichoderma viride*	63.58	78.12 b	CA2
*Trichoderma harzianum*	63.58	99.14 c	CA2
*Trichoderma atroviride*	41.03	51.82 a	CA1

* Mean values (10 repetitions) followed by different letters indicate significant differences at *p* < 0.05. ** Badalyan rating scale [30]: CA2 = complete growth of one colony upon another after remote arrest; CA1 = partial growth of one colony over another after contact arrest.

## Data Availability

Data are available by e-mail on reasonable request.

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
