# Peer review of "Trunk Injection Delivery of Biocontrol Strains of Trichoderma spp. Effectively Suppresses Nut Rot by Gnomoniopsis castaneae in Chestnut (Castanea sativa Mill.)"

_biology, 2024, doi:10.3390/biology13030143_

Round 1
Reviewer 1 Report
Comments and Suggestions for Authors
Comments
The manuscript titled "Trunk injection delivery of biocontrol strains of Trichoderma spp. effectively suppresses nut rot by Gnomoniopsis castaneae in chestnut (Castanea sativa Mill.)" has been thoroughly reviewed. The manuscript is well-written, and the work conducted by the authors represents a significant contribution to the field of biopesticides. However, there are some minor mistakes in the manuscript that the authors need to rectify. The language and overall structure of the manuscript are commendable, but there are a few grammatical mistakes that need correction. Additionally, the authors mentioned molecular characterization of Trichoderma species in the methodology section, but this was not addressed in the results section. It is crucial for the authors to provide the necessary information and findings related to the molecular characterization to maintain coherence in their work. Furthermore, I recommend that the authors consider characterizing Gnomoniopsis castaneae, the pathogen targeted by the biocontrol agents. A detailed characterization of the pathogen will enhance the comprehensiveness of the study and provide a more holistic understanding of the interactions between the biocontrol agents and the pathogen. Once these issues are addressed and the suggested improvements are incorporated, I believe the manuscript will be well-rounded and suitable for publication. The authors' efforts in investigating trunk injection delivery of biocontrol strains of Trichoderma spp. as a method to suppress nut rot are commendable, and the study holds promise for practical applications in the field of chestnut cultivation.
Minor mistakes
Line 77: Add a space between (.) and "The."
Lines 105-106: It should be "production of."
Line 101-108: I suggest authors to break this sentence in two or three sentences. I suggest following changes:
Indeed, biocontrol enables the effective and environmentally friendly control of plant diseases by harnessing the natural ability of certain microorganisms to antagonize phytopathogens through various mechanisms. These mechanisms include competition for space and nutrients, direct parasitism, production of antimicrobial compounds, and interaction with the plant system. This interaction enhances host growth and resistance by activating both ISR (Induced Systemic Resistance) and SAR (Systemic Acquired Resistance) pathways.
Lines 205-206: Rephrase the sentence to make it clear to the readers.
Line 226: I request that the authors go to the insert tab, then click on the equations tab and write the given formula using an appropriate equation that fits their present formula.
Lines 228-232: Authors should make a separate paragraph after "in the control plate" as the initial part in lines 228 and 229 are related to the given formula, and the rest should be in a separate paragraph. Also, authors should write it as: "where RM is the radius of the G. castaneae colony in the control plate, and rm is the radius of the colonies in the direction of the antagonist to follow the sequence of the formula."
Line 246: Authors have used "mL" in some places and "ml" in others. I request that the authors follow only one, not both, and use the one suggested in the author guidelines.
Line 309: Add a comma (,) after "96 hours" and "8 days."
Comments on the Quality of English LanguageThe English language of the manuscript is ok. But there very minor grammatical and punctuation mistakes which need to be corrected.
Author Response
REVIEWER 1
(in bold our responses)
The manuscript titled "Trunk injection delivery of biocontrol strains of Trichoderma spp. effectively suppresses nut rot by Gnomoniopsis castaneae in chestnut (Castanea sativa Mill.)" has been thoroughly reviewed. The manuscript is well-written, and the work conducted by the authors represents a significant contribution to the field of biopesticides. However, there are some minor mistakes in the manuscript that the authors need to rectify. The language and overall structure of the manuscript are commendable, but there are a few grammatical mistakes that need correction. Additionally, the authors mentioned molecular characterization of Trichoderma species in the methodology section, but this was not addressed in the results section. It is crucial for the authors to provide the necessary information and findings related to the molecular characterization to maintain coherence in their work. Furthermore, I recommend that the authors consider characterizing Gnomoniopsis castaneae, the pathogen targeted by the biocontrol agents. A detailed characterization of the pathogen will enhance the comprehensiveness of the study and provide a more holistic understanding of the interactions between the biocontrol agents and the pathogen. Once these issues are addressed and the suggested improvements are incorporated, I believe the manuscript will be well-rounded and suitable for publication. The authors' efforts in investigating trunk injection delivery of biocontrol strains of Trichoderma spp. as a method to suppress nut rot are commendable, and the study holds promise for practical applications in the field of chestnut cultivation.
Thank you for your appreciation of our work. We have included the characterization of both the pathogen and the Trichoderma spp.
Line 77: Add a space between (.) and "The."
Done.
Lines 105-106: It should be "production of."
Done.
Line 101-108: I suggest authors to break this sentence in two or three sentences. I suggest following changes:
Indeed, biocontrol enables the effective and environmentally friendly control of plant diseases by harnessing the natural ability of certain microorganisms to antagonize phytopathogens through various mechanisms. These mechanisms include competition for space and nutrients, direct parasitism, production of antimicrobial compounds, and interaction with the plant system. This interaction enhances host growth and resistance by activating both ISR (Induced Systemic Resistance) and SAR (Systemic Acquired Resistance) pathways.
Done.
Lines 205-206: Rephrase the sentence to make it clear to the readers.
Done.
Line 226: I request that the authors go to the insert tab, then click on the equations tab and write the given formula using an appropriate equation that fits their present formula.
Done.
Lines 228-232: Authors should make a separate paragraph after "in the control plate" as the initial part in lines 228 and 229 are related to the given formula, and the rest should be in a separate paragraph. Also, authors should write it as: "where RM is the radius of the G. castaneae colony in the control plate, and rm is the radius of the colonies in the direction of the antagonist to follow the sequence of the formula."
Done.
Line 246: Authors have used "mL" in some places and "ml" in others. I request that the authors follow only one, not both, and use the one suggested in the author guidelines.
Done.
Line 309: Add a comma (,) after "96 hours" and "8 days."
Done. We split the sentence.

Reviewer 2 Report
Comments and Suggestions for Authors
Dear Editors and Authors,
I am pleased to inform you that I have completed the review of the manuscript with the full title: “Trunk injection delivery of biocontrol strains of Trichoderma spp. effectively suppresses nut rot by Gnomoniopsis castaneae in chestnut (Castanea sativa Mill.)” (manuscript ID: biology-2874490) by Alessandra Benigno, Chiara Aglietti and Salvatore Moricca.
The occurrence of Gnomoniopsis castaneae on nuts has increased in the last ten years both in chestnut plantations in Europe (especially in Italy and Switzerland) and in Australia. The use of synthetic substances in order to protect plants has a negative impact on the environment as it leads to numerous ecological problems due to the effects on non-target organisms and favours the emergence of resistant strains of pathogens to the fungicide used. Given the ban on the use of fungicides in forests, the alternative solution against G. castanacea investigated and presented in this manuscript is of great importance.
In this study, the efficacy of the fungal genus Trichoderma (T. viride, T. harzianum and T. atroviride) in inhibiting the pathogen’s mycelial growth of G. castaneae was investigated in vitro.
In vivo experiments were conducted to investigate the ability of Trichoderma species to control the walnut blight pathogen G. castaneae. In June 2020 and 2021, were performed endotherapy treatments with 108 conidia per 1 ml of water with selected BCA strains on 66 adult chestnut trees in the Ribuio and Bandina plots.
The results of the in vivo experiments showed that the reduction in the incidence of G. castaneae between treated and untreated stands in the Ribuio and Bandina groves was 22% and 26% in 2020 and 29% and 40% in 2021. The fact that in the second year a greater reduction in the incidence of nut fruits was achieved means that there was an cumulative effect during two years treatments.
I congratulate you on your work, a very topical and interesting subject, the results of which can be applied not only to the protection of chestnuts and other nut species against Gnomoniopsis castaneae, but also to other plant species and their pathogens. I think the introduction is well written. The results of the investigation are adequately presented and the discussion is good enough. Since the authors have data on the age of all 66 chestnut trees treated in vivo at the Ribuio and Bandina sites, I would like to ask them to consider whether the age of the plants can be linked to the results obtained from the endotherapeutic treatments.
Thank you for inviting me to participate in the journal's manuscript review process. As the authors will see from the comments below, this manuscript may become acceptable for publication in Biology after minor revision.
Kind regards,
After review, I suggest changes to the manuscript and offer the following comments:
Line 187: Please complete the existing text with the details of the manufacturer of streptomycin sulfate (add the city).
Line 201: Please add the details of the manufacturer of the Genomic DNA Miniprep extrac-200 tion kit (city, state).
Line 215: Please add the details of the manufacturer and respect the uniformity of the text (manufacturer, city, state).
Line 261: Please replace the lower case letter of the word "fig. 2." with capital letter.
Lines 280-281: The full data for MEA is not necessary as it is already written in the previous chapters. Please replace the text “Petri dishes filled with 2% Malt Extract Agar (MEA, DIFCO, Detroit, Michigan, USA)” with “Petri dishes filled with 2% MEA”.
Linе 311: You are referring to figure number 3 and not number 1. Please correct the number.
Line 313: Please replace the number 1 with the number 3.
Lines 324-325: Please replace the text “a) Trichoderma viride/G. castaneae interaction; b) Trichoderma harzianum/G. castaneae interaction; and c) Trichoderma atroviride/G. castaneae interaction” with the text “a) T. viride/G. castaneae interaction; b) T. harzianum/G. castaneae interaction; and c) T. atroviride/G. castaneae interaction.”
Lines 328-329: Please replace the text “Antagonism of Trichoderma viride, Trichoderma harzianum and Trichoderma atrovirid” with the text “Antagonism of T. viride, T. harzianum and T. atroviride”.
Lines 347: Please replace the text “of the target pathogen was 11% and 23% in the treated plots and 40% and 63% in the” with the text “of the target pathogen was 23% and 11% in the treated plots and 63% and 40% in the”.
Line 349: Please replace the text “resulted as 22% and 26% in 2020 and 29% and 40% in 2021 (Figg. 4-5)” with the text “resulted as 26% and 22% in 2020 and 40% and 29% in 2021 (Fig. 5).
Lines 382-400: Whether the results achieved with endotherapeutic treatments can be linked to the different ages of the chestnut plants in the areas studied?
Author Response
REVIEWER 2
(in bold our responses)
I am pleased to inform you that I have completed the review of the manuscript with the full title: “Trunk injection delivery of biocontrol strains of Trichoderma spp. effectively suppresses nut rot by Gnomoniopsis castaneae in chestnut (Castanea sativa Mill.)” (manuscript ID: biology-2874490) by Alessandra Benigno, Chiara Aglietti and Salvatore Moricca.
The occurrence of Gnomoniopsis castaneae on nuts has increased in the last ten years both in chestnut plantations in Europe (especially in Italy and Switzerland) and in Australia. The use of synthetic substances in order to protect plants has a negative impact on the environment as it leads to numerous ecological problems due to the effects on non-target organisms and favours the emergence of resistant strains of pathogens to the fungicide used. Given the ban on the use of fungicides in forests, the alternative solution against G. castaneae investigated and presented in this manuscript is of great importance.
In this study, the efficacy of the fungal genus Trichoderma (T. viride, T. harzianum and T. atroviride) in inhibiting the pathogen’s mycelial growth of G. castaneae was investigated in vitro.
In vivo experiments were conducted to investigate the ability of Trichoderma species to control the walnut blight pathogen G. castaneae. In June 2020 and 2021, were performed endotherapy treatments with 108 conidia per 1 ml of water with selected BCA strains on 66 adult chestnut trees in the Ribuio and Bandina plots.
The results of the in vivo experiments showed that the reduction in the incidence of G. castaneae between treated and untreated stands in the Ribuio and Bandina groves was 22% and 26% in 2020 and 29% and 40% in 2021. The fact that in the second year a greater reduction in the incidence of nut fruits was achieved means that there was an cumulative effect during two years treatments.
I congratulate you on your work, a very topical and interesting subject, the results of which can be applied not only to the protection of chestnuts and other nut species against Gnomoniopsis castaneae, but also to other plant species and their pathogens. I think the introduction is well written. The results of the investigation are adequately presented and the discussion is good enough. Since the authors have data on the age of all 66 chestnut trees treated in vivo at the Ribuio and Bandina sites, I would like to ask them to consider whether the age of the plants can be linked to the results obtained from the endotherapeutic treatments.
Thank you for inviting me to participate in the journal's manuscript review process. As the authors will see from the comments below, this manuscript may become acceptable for publication in Biology after minor revision.
Thank you for your appreciation of our work. Regarding your question (which we consider very interesting and worthy of deeper investigation) on the age of the plants, we have treated plants with ages ranging from less than 30 to more than 150 years (Table 1) but we found no differences in the response to endotherapic treatments. A targeted future study that includes replicates with blocks of different age would be interesting.
Line 187: Please complete the existing text with the details of the manufacturer of streptomycin sulfate (add the city).
Done.
Line 201: Please add the details of the manufacturer of the Genomic DNA Miniprep extraction kit (city, state).
Done.
Line 215: Please add the details of the manufacturer and respect the uniformity of the text (manufacturer, city, state).
Done.
Line 261: Please replace the lower case letter of the word "fig. 2." with capital letter.
Done.
Lines 280-281: The full data for MEA is not necessary as it is already written in the previous chapters. Please replace the text “Petri dishes filled with 2% Malt Extract Agar (MEA, DIFCO, Detroit, Michigan, USA)” with “Petri dishes filled with 2% MEA”.
Done.
Linе 311: You are referring to figure number 3 and not number 1. Please correct the number.
Done.
Line 313: Please replace the number 1 with the number 3.
Done.
Lines 324-325: Please replace the text “a) Trichoderma viride/G. castaneae interaction; b) Trichoderma harzianum/G. castaneae interaction; and c) Trichoderma atroviride/G. castaneae interaction” with the text “a) T. viride/G. castaneae interaction; b) T. harzianum/G. castaneae interaction; and c) T. atroviride/G. castaneae interaction.”
Done.
Lines 328-329: Please replace the text “Antagonism of Trichoderma viride, Trichoderma harzianum and Trichoderma atroviride” with the text “Antagonism of T. viride, T. harzianum and T. atroviride”.
Done.
Lines 347: Please replace the text “of the target pathogen was 11% and 23% in the treated plots and 40% and 63% in the” with the text “of the target pathogen was 23% and 11% in the treated plots and 63% and 40% in the”.
Done.
Line 349: Please replace the text “resulted as 22% and 26% in 2020 and 29% and 40% in 2021 (Figg. 4-5)” with the text “resulted as 26% and 22% in 2020 and 40% and 29% in 2021 (Fig. 5).
Done.
Lines 382-400: Whether the results achieved with endotherapeutic treatments can be linked to the different ages of the chestnut plants in the areas studied?
Done. We answered this above.

Round 2
Reviewer 1 Report
Comments and Suggestions for Authors
The authors have addressed all the previous comments and can be accepted for publication in Biology after completing following minor error:
1. Authors have used * in the formula. I request authors to go the insert tab, then click on symbols and used this symbol (×).